# Xenoestrogen concentration in women with endometriosis or leiomyomas: A case-control study

Victoria Valdes-Devesa[1,2]* , David Sanz-Rosa[1] , Israel J. Thuissard-Vasallo[1‡], Cristina Andreu-Vázquez[1‡], Ricardo Sainz de la Cuesta[1,2]

**1** Department of Medicine, Faculty of Biomedical and Health Sciences, Universidad Europea de Madrid, Villaviciosa de Odon, Madrid, Spain, **2** Department of Obstetrics & Gynecology, Hospital Universitario Quironsalud Madrid, Pozuelo de Alarcon, Madrid, Spain

☯ These authors contributed equally to this work.
‡ IJTV and CAV also contributed equally to this work.
* victoria25valdes@hotmail.com, victoria.valdes@quironsalud.es

**Data Availability Statement:** All relevant data are within the manuscript and its Supporting Information files, named database and database encoding.

## Abstract

### Background

Xenoestrogens are synthetic or naturally occurring chemicals capable of altering the endocrine system of humans and animals owing to their molecular similarity to endogenous hormones. There is limited data regarding their effects on women´s health. Chronic exposure to xenoestrogens can promote the development of estrogen-related diseases.

### Objectives

To examine xenoestrogen concentration (TEXB-α) differences between women with leiomyomas or endometriosis and control women, and to study the relationship between the clinical and sociodemographic characteristics of these patients and their xenoestrogen levels.

### Methods

Prospective case-control study. We selected 221 women who underwent surgery at Quironsalud Madrid University Hospital between 2017 and 2021. The cases included 117 patients: 74 women who underwent surgery for uterine leiomyomas, 21 with endometriosis, and 22 with both pathologies. The control group comprised 104 healthy women who underwent surgical procedures for other reasons. TEXB-α was determined in the omental fat of all patients. Using a questionnaire and reviewing the patients' medical records, we collected sociodemographic data and other relevant variables.

### Results

A significant majority of study participants (68.8%) had detectable levels of xenoestrogens. We found no association between TEXB-α levels in omental fat and the presence of myomas or endometriosis. In the case group, women living or working in Madrid Community

**Funding:** The XENOBEM study received the National Grant for Clinical Research from the Fundacion Dexeus Mujer in 2019 (grant number B26/18). The grant was used to fund the processing of part of the study's samples. No external peer review was included. The funder did not participate in the research or subsequent publication of the results.

**Competing interests:** The authors have declared that no competing interests exist.

exhibited, on average, 3.12 Eeq pM/g higher levels of TEXB-α compared to those working in other areas (p = 0.030). Women who referred to the use of estrogen-containing hormonal contraceptives had, on average, 3.02 Eeq pM/g higher levels of TEXB-α than those who had never used them (p = 0.022).

## Conclusions

This study found no association between omental xenoestrogen levels and leiomyomas or endometriosis. However, their presence in most participants and their association with highly polluted areas emphasizes the importance of limiting environmental exposure to these substances. We also identified an association between hormonal contraceptive use and xenoestrogen concentration.

## Introduction

Endocrine-disrupting chemicals (EDCs) are exogenous substances that alter the functions of the endocrine system, resulting in adverse health effects in an intact organism, its progeny or subpopulations [1]. Xenoestrogens, including phytoestrogens, phthalates, polychlorinated byphenils, isoflavonoids, parabens, bisphenol A, and other exogenous estrogens, can act as EDCs.

Environmental exposure to EDCs is widespread [1]. Due to their lipophilic characteristics and slow metabolism and detoxification, they accumulate in adipose tissue, where they are released into the bloodstream [1]. The pathogenic mechanisms of xenobiotics are multiple and complex; they can affect the synthesis, release, or transport of natural hormones or modulate the response of target tissues. They can also exert stimulant or inhibitory effects depending on their concentration. Interaction with other EDCs can result in agonistic or antagonistic effects, depending on the composition of the mixture and the concentration of each substance [2]. Although EDCs can affect the endocrine system at any stage of life, embryologic or fetal stages are critical because they can cause epigenetic modifications that lead to morbidity in adult life [2].

Epidemiological studies and experimental models have reported a potential link between xenoestrogens and hormone-dependent pathologies, such as leiomyomas or endometriosis [3–12]. However, a direct causal relationship between these entities has yet to be established.

Most publications have focused on individual or small groups of EDCs and their effects on health outcomes [3–5], failing to consider the agonistic, antagonistic, or synergistic interactions between multiple xenobiotics that can modify the final estrogenic effects of a mixture [2]. Furthermore, the potential association between individual xenoestrogens and certain diseases should be assessed by considering their interactions with other EDCs [13]. In accordance with this idea, the assessment of the total concentration of xenoestrogens has emerged as a reliable biomarker for the cumulative effect of xenoestrogen mixtures on organisms [1].

Additionally, most publications have reported the use of blood, urine, or even subcutaneous fatty tissue to quantify xenoestrogens [6–8]. Adipose tissue has been regarded as reliable for evaluating the long-term bioaccumulation of xenoestrogens [2]. However, visceral fat has shown a higher deleterious metabolic effect than subcutaneous fat [14] and seems to be a more appropriate tissue for this purpose.

The XENOBEM (XENOBiotics, Endometriosis and Myomas) study aimed to determine whether there is an association between xenoestrogens and the presence of leiomyomas or

endometriosis, and to assess the xenoestrogen concentration in omental fat. We conducted a case-control pilot study to compare women who underwent surgery and had leiomyomas or endometriosis with women who went through surgery for other benign indications.

## Material and methods

The present study was approved by the Institutional Ethics Review Board of Fundacion Jimenez Diaz Hospital (Madrid, Spain) on 12/20/2016, reference number 21/2016.

### Objectives

The main objective of the XENOBEM study was to determine whether there is a positive association between the alpha fraction of the total xenoestrogen burden (TEXB-α) present in omental fat and the presence of endometriosis or uterine leiomyomas. Secondary endpoints included the relationship between the clinical and sociodemographic characteristics of these patients and their xenoestrogen levels.

### Main outcome measure

Xenoestrogen concentration (TEXB-α) in omental fat.

### Study population

The XENOBEM study is a case-control prospective study that included patients who underwent abdominal surgery at the University Hospital Quironsalud Madrid between February 24, 2017, and February 1, 2021. The inclusion criteria for the case group were women with benign leiomyomas or endometriosis, while the control group included women with other non-hormonal benign diseases (tubo-ovarian abscesses secondary to a pelvic inflammatory disease or hemorrhagic ovarian follicles that underwent surgery) or pregnant women who needed a cesarean section. The exclusion criteria for the case group included intraoperative findings or anatomopathological results indicating conditions other than leiomyomas or endometriosis. Women in the control group with a history of fibroids or endometriosis were excluded.

All patients signed an informed consent form before participating in the study and completed a questionnaire that focused on clinical and sociodemographic data.

### Variables of study

We created a coded database including TEXB levels, age, body mass index (kg/m$^2$), medical and surgical history, parity, use of hormonal contraceptives (HC), menopausal status, use of hormonal therapy for menopause, smoking status, family history of fibroids, endometriosis, or gynecological cancer, and the main place of residence and work in the last 10 years. In the case group, we also analyzed the presence and type of symptoms, location, size, and number of fibroids, type of endometriosis, and bowel involvement.

### Biochemical analyses

During surgery, we collected a 2–4 g omentum sample that was coded and stored at -80˚C until chemical analysis. Bioaccumulative compounds were extracted according to the method described by Fernandez et al. [7] and separated by high-performance liquid chromatography (HPLC). This technique allows the separation of exogenous hormones (the so-called alpha fraction) from the beta fraction, composed of endogenous estrogens and more polar xenoestrogens that behave as natural hormones, being quickly metabolised and not acting as EDCs [6]. Subsequently, the alpha fraction was tested in the E-screen bioassay for estrogenicity,

following the technique originally described by Soto et al. [9] with minor modifications [10, 11]. Each sample was assayed in triplicate with a negative (steroid-free) control and a positive control, where 17-β-estradiol was added to a culture of MCF-7 breast cancer cells [7] at a concentration that would induce a maximum proliferative effect. The proliferative effect was transformed into estradiol equivalent units (Eeq) by reading the dose-response curve. The results were expressed as the alpha Total Effective Xenoestrogen Burden (TEXB-α) in Eeq per gram of lipid. TEXB-α is considered a reliable biomarker for the combined effect of mixtures of halogenated organic lipophilic xenoestrogens [6] and is also referred to in this article as the xenoestrogen concentration.

The minimum concentration of TEXB-α required to elicit a significant proliferative effect on the MCF-7 cell culture, (as compared to the steroid-free control culture), was determined to be 0.1 Eeq pM/g, which represents the limit of detection (LD) for the bioassay [1].

## Statistical analysis

The descriptive analysis included the clinical and sociodemographic variables of the study. We used absolute (n) and relative (%) frequencies for qualitative variables and mean ± standard deviation (SD) or median and interquartile range (IQR) for quantitative variables depending on their normality. Categorical variables were summarized using absolute and relative frequencies, and quantitative variables using means and standard deviations or medians and interquartile ranges based on the normality of the variable.

To assess differences in xenoestrogen concentrations between the case and control groups, we used Student's t-test or Mann-Whitney U test. We employed linear regression anaylsis to determine the influence of the study group and the impact of clinical and sociodemographic variables on xenoestrogen levels.

We used chi-square tests to assess differences in the proportion of detectability between groups and other variables of interest.

Furthermore, we conducted subgroup analyses to compare the detectability and xenoestrogen concentrations between controls and women with fibroids, endometriosis, or both.

Additionally, we performed all comparisons between cases and controls regarding xenoestrogen levels after converting the concentration values to their natural logarithms.

Following the methodology described in previous publications [7], we grouped participants into tertiles of serum TEXB-α based on TEXB-α distribution among controls and compared the variables of interest among tertiles.

To ensure that the results obtained in the comparison between the case and control groups as a whole were not affected by the mixture of pregnant and non-pregnant individuals in both groups, we conducted a second comparison: in order to homogenize the groups, we compared xenoestrogen concentrations including only pregnant women in the control group. In the case group, pregnant women were excluded. In summary, we performed additional analyses comparing TEXB-α levels in a control group that included only pregnant women with those in a case group conformed by non-pregnant patients diagnosed with leiomyomas or endometriosis.

Statistical significance was set at 95% (p≤0.05). SPSS software (version 29.0; IBM, Armonk, NY) was used for all statistical analyses.

## Results

Out of the 240 omental fat samples initially collected, 11 were not analyzed because of anatomopathological findings other than fibroids, endometriosis, or incorrect labelling or preservation. Of the remaining 229 samples, 117 corresponded to the case group: 74 patients with

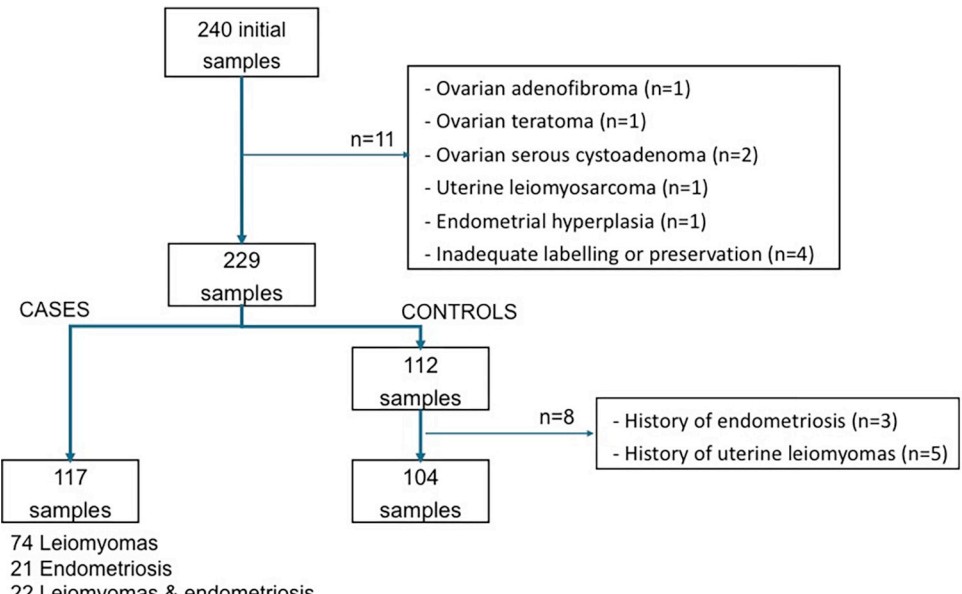

**Fig 1. Visual representation of the study sample enrollment flow.**

leiomyomas, 21 with endometriosis, and 22 with both leiomyomas and endometriosis. Of these 117 patients, eight (6.8%) were pregnant women with miomas/endometriosis undergoing cesarean section. Within the control group, we excluded eight patients due to a past medical history of one of these diseases, resulting in a total of 104 patients (Fig 1). In the control group, 93 (86.5%) of the 104 patients were pregnant women who underwent cesarean sections for obstetric indications.

## Main characteristics of the study population

Table 1 presents the main characteristics of the groups. A family history of fibroids was more common in the case group than in the control group (49 vs. 24 cases, p = 0,006).

Both groups had a higher percentage of patients living in the Madrid Community: 85 women (83.3%) in the control group and 78 in the case group (67.2%) (p = 0.006). In addition, 87 women in the control group (85.3%) and 79 women among the cases (68.1%) had workplaces within the Madrid Community (p = 0.003).

Regarding the clinical features of the case group, we should point out that a great number of patients had multiple fibroids and in different uterine locations. In addition, the majority of patients with endometriosis had a combination of ovarian and deep endometriosis. The main indications for surgery in patients with leiomyomas were abnormal uterine bleeding, and pelvic pain for women with endometriosis, both unresponsive to medical treatment.

## Omentum levels of TEXB-α

The detection frequency of TEXB-α for the samples was 68.8%, with no difference between the groups: 76 of the 104 control women (73.1%) and 76 of the 117 cases (65.0%) had xenoestrogen levels over the LD of 0.1 Eeq pM/g (p = 0.194). There were also no differences in xenoestrogen detection frequency among the different subgroups: 49 of 74 patients with leiomyomas (66.2%), 13 of 21 women with endometriosis (61.9%), and 14 of 22 patients with both pathologies (63.6%) had detectable xenoestrogen levels (p = 0.603).

**Table 1. Main sociodemographic and clinical characteristics of the study population.**

| Main characteristics | Controls (n = 104) | Cases (n = 117) | p value |
|---|---|---|---|
| **Age (years)** | 37.00 [6.00] | 43.00 [9.00] | **< 0.001** |
| **BMI\* (kg/m²)** | 23.72 [5.58] | 23.80 [5.76] | 0.234 |
| Underweight (<18,5) | 4 (3.85) | 5 (4.27) | NA |
| Normal weight (18,5–24,9) | 61 (58.65) | 62 (52.99) | NA |
| Overweight(25–29,9) | 29 (27.88) | 29 (24.79) | NA |
| Obesity (>30) | 10 (9.62) | 21 (17.95) | NA |
| **Pathology** | | | |
| No hormone-dependent pathology | 104 (100.00) | - | |
| Leiomyomas | - | 74 (63.25) | |
| Endometriosis | - | 21 (17.95) | |
| Leiomyomas and endometriosis | - | 22 (18.80) | |
| **Hormonal contraception (HC)** | | | |
| Ever users of HC | 60 (57.69) | 62 (53.45) | 0.527 |
| Oral contraceptives | 51 (49.04) | 57 (50.00) | 0.887 |
| Transdermal patch | 4 (3.85) | 3 (2.63) | 0.712 |
| Vaginal ring | 13 (12.50) | 6 (5.26) | 0.058 |
| LNG releasing IUD | 3 (2.88) | 10 (8.77) | 0.067 |
| Contraceptive implant | 2 (1.92) | 2 (1.75) | 1.000 |
| **Time of HC use (years)** | 6.00 [7.00] | 5.00 [7.75] | 0,737 |
| **Parity** | | | |
| Number of children | 2 [1.00] | 1 [2.00] | **< 0,001** |
| Nuliparous | 3 (2.88) | 51 (43.97) | **< 0,001** |
| Age at first childbirth (years) | 33.00 [6.00] | 30.0 [9.00] | **0,004** |
| **Postmenopausal** | 2 (1.92) | 6 (5.17) | 0,286 |
| **Current smokers** | | | |
| Yes | 5 (4.81) | 18 (15.52) | **0,010** |
| Years of smoking habit | 10.80 [10.03] | 22.50 [9.91] | **0,043** |
| Number of cigarretes/day | 10.00 [9.00] | 10.00 [9.75] | 0,880 |
| No | | | |
| Former smokers | 47 (45.19) | 41 (35.34) | 0,137 |
| Years since termination | 3.00 [7.00] | 9.50 [11.75] | **<0,001** |
| Never smokers | 52 (50.00) | 57 (49.14) | 0,898 |

Considering only those patients with xenoestrogen concentrations over the LD, the median level for controls was 4.91 [5.81], vs 4.62 [4.34] Eeq pM/g in cases (p = 0.078). We observed a significant dispersion of the values for both groups, as shown in Fig 2. We found no differences in the TEXB-α levels between controls and patients with leiomyomas, endometriosis, or both: patients with fibroids showed TEXB-α levels of 4.63 [3.77] Eeq pM/g; for those with endometriosis, the median was 4.63 [7.32] Eeq pM/g, while for patients with both endometriosis and leiomyomas, the median concentration was 3.35 [7.28] Eeq pM/g.

Comparisons made after the logarithmic conversion of xenoestrogen concentrations showed no statistically significant differences between the two groups.

Finally, we compared xenoestrogen levels after homogenization of the groups. Thus, the control group, consisting of 93 pregnant women, was compared with 109 non-pregnant cases. This additional analysis confirmed the absence of statistically significant differences in the detectability and TEXB-α levels between the groups as well as the dispersion of the values.

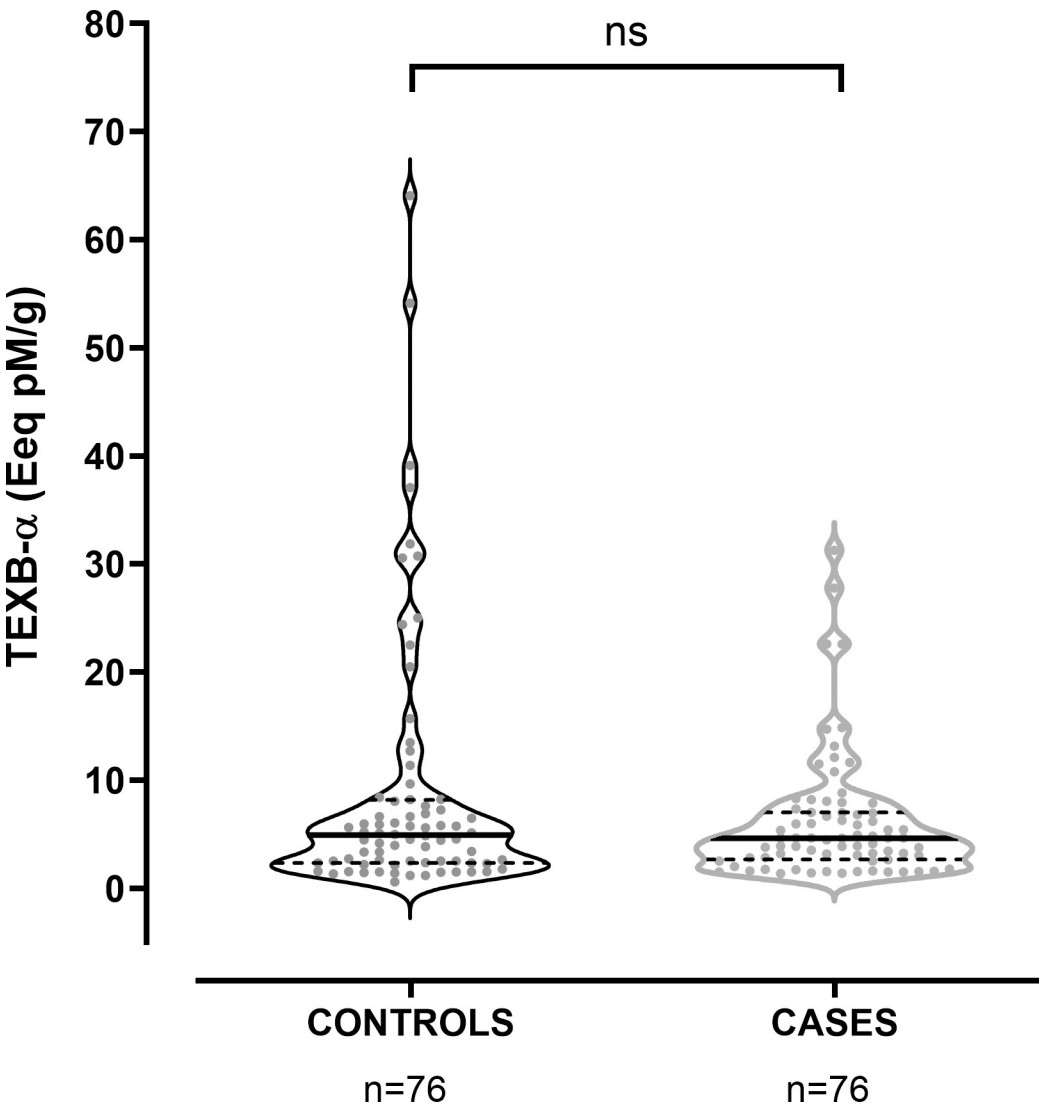

**Fig 2. Distribution of detectable TEXB-α levels in cases and controls.** The solid and dashed lines show the median and quartile values, respectively.

## Relationship between xenoestrogen levels and clinical and sociodemographic characteristics

We performed a comparative analysis that included only patients with TEXB-α levels above the limit of detection. The results were similar to those obtained from the comparative study that included all case and control groups (S1 Table). The controls were significantly younger (36.0 [6.0] vs 43.0 [10.0] years). In addition, cases were more likely to be ever-users of levonor-gestrel-releasing intrauterine devices (LNG-IUDs) and reported more years of smoking; the former smokers in the group of cases had been smoke-free for more years than controls. Both the case and control groups had a higher percentage of women living within the Madrid Community: 61 patients (82.4%) in the control group and 50 (65.8%) in the case group (p = 0.020). In addition, 62 women in the control group (83.8%) and 52 among the cases (68.4%) had workplaces in the Madrid Community (p = 0.028).

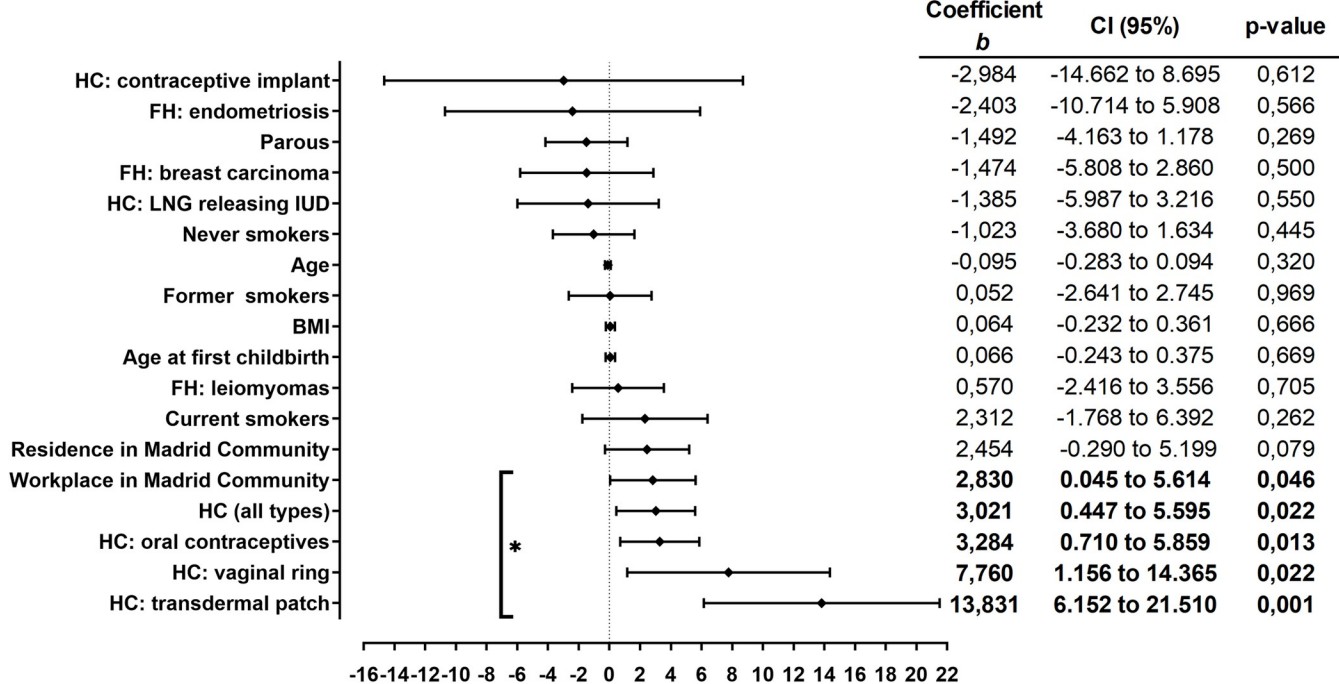

| | Coefficient b | CI (95%) | p-value |
|---|---|---|---|
| HC: contraceptive implant | -2,984 | -14.662 to 8.695 | 0,612 |
| FH: endometriosis | -2,403 | -10.714 to 5.908 | 0,566 |
| Parous | -1,492 | -4.163 to 1.178 | 0,269 |
| FH: breast carcinoma | -1,474 | -5.808 to 2.860 | 0,500 |
| HC: LNG releasing IUD | -1,385 | -5.987 to 3.216 | 0,550 |
| Never smokers | -1,023 | -3.680 to 1.634 | 0,445 |
| Age | -0,095 | -0.283 to 0.094 | 0,320 |
| Former smokers | 0,052 | -2.641 to 2.745 | 0,969 |
| BMI | 0,064 | -0.232 to 0.361 | 0,666 |
| Age at first childbirth | 0,066 | -0.243 to 0.375 | 0,669 |
| FH: leiomyomas | 0,570 | -2.416 to 3.556 | 0,705 |
| Current smokers | 2,312 | -1.768 to 6.392 | 0,262 |
| Residence in Madrid Community | 2,454 | -0.290 to 5.199 | 0,079 |
| Workplace in Madrid Community | **2,830** | **0.045 to 5.614** | **0,046** |
| HC (all types) | **3,021** | **0.447 to 5.595** | **0,022** |
| HC: oral contraceptives | **3,284** | **0.710 to 5.859** | **0,013** |
| HC: vaginal ring | **7,760** | **1.156 to 14.365** | **0,022** |
| HC: transdermal patch | **13,831** | **6.152 to 21.510** | **0,001** |

**Fig 3. Simple linear regression analysis of the cases.** The regression coefficients and 95% confidence intervals are shown in the right-hand panel. BMI, body mass index; FH, Family history; HC, Hormonal contraceptives; LNG, Levonorgestrel; IUD, intrauterine device. * p<0.05.

None of the study variables showed to have an influence on the xenoestrogen concentration. Nevertheless, when we analyzed the case and control groups separately, we found that in the group of cases, the use of hormonal contraceptives and having a residence or workplace within the Madrid Community were associated with higher TEXB-α levels (Figs 3 and 4). Interestingly, patients who reported living or working mainly in Madrid during the last 10 years had, on average, 3.12 Eeq pM/g TEXB-α higher levels than those working outside the area (p = 0.030). Women who reported having ever used hormonal contraceptives had, on average, 3.02 Eeq pM/g higher levels than those who had never used these drugs (p = 0.022).

Notably, the time between stopping contraceptives and undergoing surgery averaged 8.0 (8.0) years. When investigating the type of contraceptive used, this difference remained for all estrogen-containing drugs but did not apply to progesterone-releasing devices, such as IUDs or contraceptive implants. We confirmed these findings in the tertile comparison (Fig 5): while there were no differences in the control group (p = 0.299), we found a significant difference in the distribution of the cases, where oral contraceptive users were mainly in T2 (2.62–6.51) and T3 (>6.51 Eeq pM/g), (p = 0.049).

The results of the case group were confirmed by multivariate analysis after adjusting for age and BMI. A relationship between the use of hormonal contraceptives and having a workplace or residence within the Madrid Community was discarded (Fig 6).

Xenoestrogen concentration showed no relationship with the clinical presentation of leiomyomas or endometriosis, such as symptomatology, fibroid size or location, endometriosis type, or bowel involvement.

## Discussion

Most patients had detectable TEXB-α levels, but there were no significant differences in xenoestrogen concentrations between cases and controls. TEXB-α levels in the omentum were

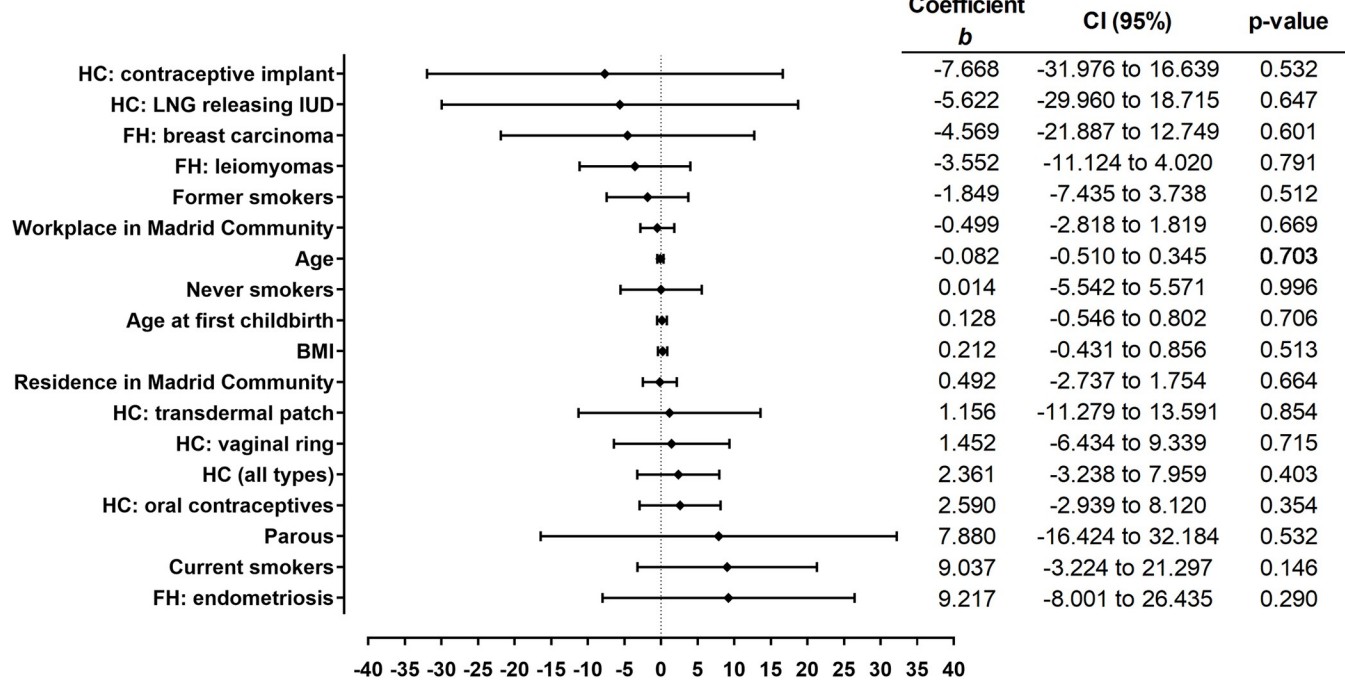

**Fig 4. Simple linear regression analysis of the controls.** The regression coefficients and 95% confidence intervals are shown in the right-hand panel. BMI, body mass index; FH, Family history; HC, Hormonal contraceptives; LNG, Levonorgestrel; IUD, intrauterine device.

not associated with the presence of leiomyomas or endometriosis. However, living or working in the Madrid Community or using estrogen-containing contraceptives were associated with higher xenoestrogen levels in the case group.

The omental samples of patients with endometriosis or leiomyomas did not have significantly higher TEXB-α levels than the control group. Taking into consideration the high percentage of patients with detectable levels of TEXB-α, this leads to the conclusion that xenoestrogens, at the concentrations detected in our study, are not associated with leiomyomas or endometriosis. Although the proliferative and carcinogenic effects of xenoestrogens have been widely reported [12], their mechanisms of action are complex and variable. Animal models have demonstrated a special type of dose-response curve, called the hormesis effect, where low doses can have a stimulatory effect, whereas high doses may be inhibitory [12, 13]. Additionally, there is a lack of data about the toxicity of mixtures of xenobiotics, and there is no known threshold above which xenoestrogens have a harmful effect on living beings. In our study, alpha-fraction levels above the detection limits were present in most patients (68.8%, similar to previous publications [6]), and a wide variability in the concentration of xenoestrogens was observed. In short, although this study demonstrated an absence of an association between xenobiotics and hormone-dependent diseases, the high percentage of patients with detectable levels of TEXB-α confirms the validity of concerns about exposure to xenobiotics that, even at very low doses, have demonstrated a negative effect on the endocrine system of humans and animals. We also lacked information regarding intrauterine exposure of the study participants, which may affect the development of hormone-dependent diseases in adulthood. Compiling intrauterine exposure data for the adult population is not feasible. Therefore, given the heightened susceptibility to EDCs during intrauterine life [2], investigating the offspring of women with known xenoestrogen concentrations during pregnancy would be of particular interest.

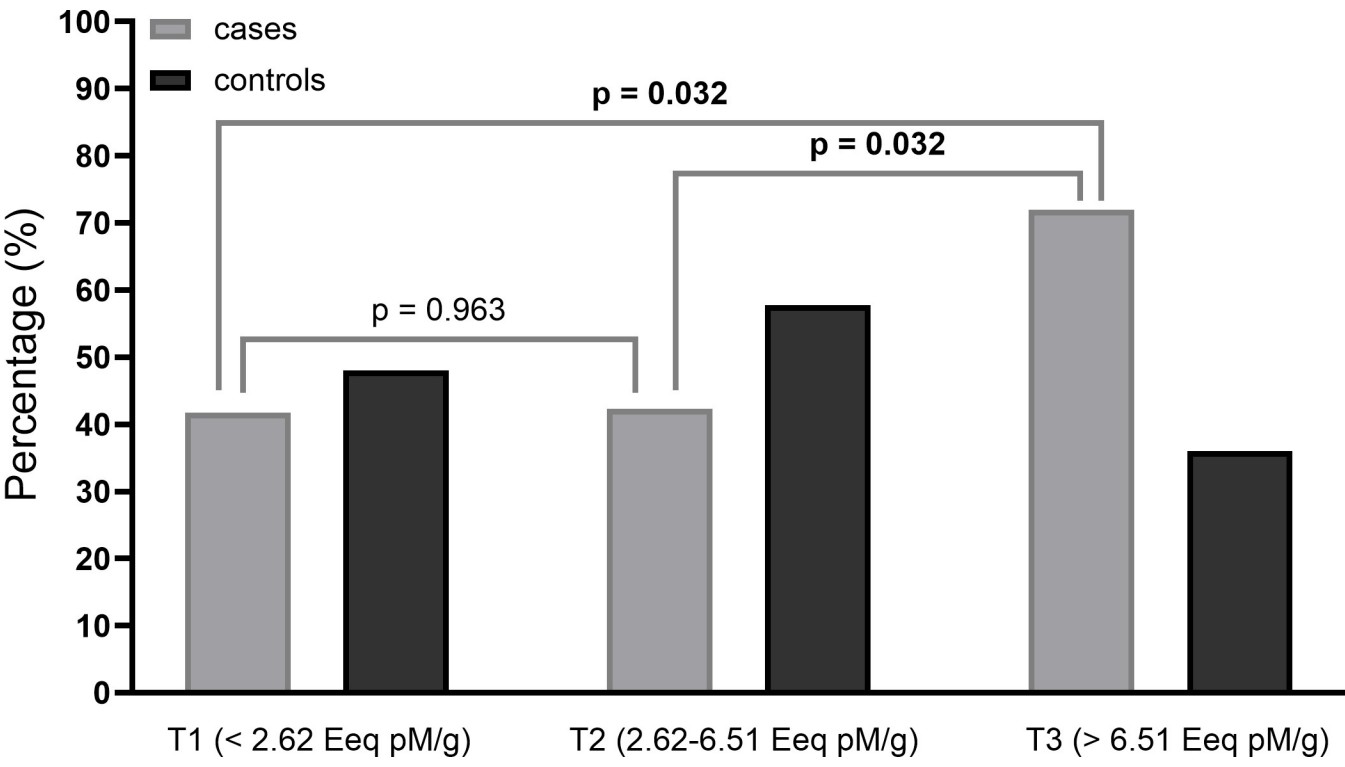

**Fig 5. Comparison of tertiles of TEXB-α among oral contraceptive users in cases and controls.**

The absence of a relationship between xenoestrogens and age is consistent with the findings described in previous publications [1, 15], although these can be quite diverse and sometimes contradictory. Our results contradict the well-known cumulative effect of endocrine disruptors, which suggests that higher levels of xenobiotics can be detected in biological tissues with advancing age. However, it is important to consider that despite the significant age difference between cases and controls, the median age of both groups differed by only six years, limiting the assessment of this cumulative effect with age. This finding aligns with the discovery described by Arrebola et al. [1], who observed an association between the levels of endogenous (but not exogenous) hormones in adipose tissue with advancing age. Fernandez et al. [16] described an inversely proportional association between age and TEXB-α levels in fatty tissues in their study population.

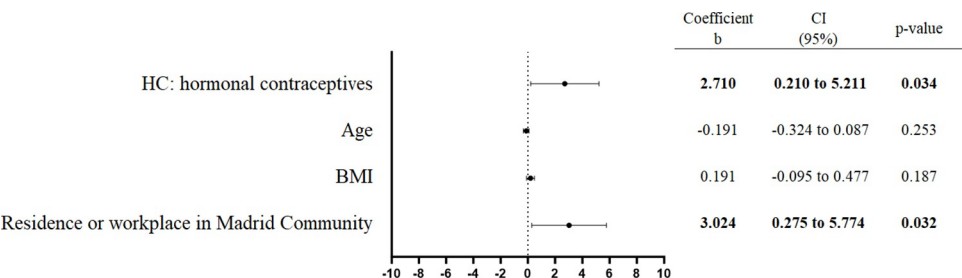

**Fig 6. Multiple regression analysis of the cases.** The regression coefficients, 95% confidence intervals (CI) and p-value are shown in the right-hand panel. BMI, body mass index.

A positive association between xenobiotics and BMI might be expected. First, because of the higher presence of fatty tissue, which acts as a reservoir for endocrine disruptors. Secondly, due to associations with certain lifestyle habits, such as sedentary behavior or consumption of processed foods, usually linked with plastic packaging, which are common sources of phthalates or bisphenol A. Despite both groups being balanced not only in body mass index but also in percentages of underweight, normal weight, overweight, and obesity, no association was observed between BMI and TEXB-α. This observation coincides with previous publications [1], which attributed it to interactions with endogenous estrogens. Other studies have identified lower levels of individual xenobiotics with higher BMI, theorizing that body mass index could have a dilutional effect on endocrine disruptors [17], although further studies are needed to determine the real relationship between xenoestrogens and BMI.

Regarding the concentrations of TEXB described in our research, it is not possible to compare with previous publications since no previous investigation fulfils the criteria of the XENOBEM study. Most of these studies have determined individual xenobiotics, but not the effect of the total xenoestrogen burden. This is the case in the ENDO study [3], which explored the relationship between endometriosis and organic pollutants quantified in omentum samples, but analyzed separately individual xenobiotics.

Other publications have measured xenobiotic concentrations in urine or serum or focused on different diseases. For instance, the publications made by Fernandez et al. [1, 7, 9, 14, 16] determined the total effective xenoestrogen burden, but the reference samples were serum or peripheral adipose tissue. Furthermore, their study did not focus on benign hormone-dependent diseases.

In the group of cases, patients who lived or worked in the Madrid Community during the last 10 years showed significantly higher levels of xenoestrogens than women with a residence or workplace in other areas. This region is composed of numerous highly polluted urban areas, while patients coming from other territories generally reported living in smaller cities or rural areas. Khomenko et al [18] reported that the metropolitan area of Madrid, including all the nearby cities surrounding the capital, is among the European areas with a higher mortality attributable to pollution. Air pollution levels in Madrid are comparable to those in other major European cities such as Antwerp, Paris, Milan and Barcelona [18]. Our findings highlight the negative impact of environmental pollution on xenobiotic absorption. Considering that the estimated cost of diseases caused by EDC has been calculated as €163 billion per year [19], the results of the XENOBEM study emphasize the necessity of stringent regulations for industrial usage.

The association between hormonal contraceptive use and xenoestrogen concentration in the cases was unexpected. Women who reported to have ever used HC containing estrogens showed an average of 3.02 Eeq pM/g above those who had never used them: considering that the median in this group was 4.62 Eeq pM/g, this results in an increase of 65.4%. Interestingly, estrogens contained in contraceptives behave as endogenous hormones, being rapidly metabolized and eliminated in urine and eluting in the beta fraction (comprised of endogenous hormones) during the HPLC fractionation process. Therefore, there must be another source of xenobiotics in oral hormonal contraceptives. It must also be noted that the mean time between completion of oral contraceptive treatment and surgery was 8 years. This finding of higher xenoestrogen levels over such an extended period is consistent with the behavior of xenobiotics, as they have demonstrated to remain in the body for prolonged periods of time owing to their slow detoxification [1].

The common use of xenobiotics in drugs has been widely described [4, 20, 21]. Some xenoestrogens, such as phthalates, are permitted as excipients, artificial flavors, or to make enteric coatings. There is a concern due to their presence in many pharmaceuticals, such as

antidepressants, antiacids, and anti-inflammatory drugs. Nevertheless, to our knowledge, the presence of xenobiotics in contraceptive pills has not been reported. The use of EDCs as plasticizers is highly prevalent, and their potential use in contraceptive blister packs, as well as in the manufacturing of contraceptive rings or patches, cannot be disregarded. To date, only one publication has reported elevated levels of phthalates among current users of vaginal contraceptive rings [22]. Confounding factors, such as educational level, social class, occupational history, and lifestyle habits, may play a role and have been shown to be associated with TEXB-α levels [8, 16, 23, 24]. In this study, we did not examine these possible confounding variables. However, as far as we know, this is the first research reporting an association between the use of HC and the total effective xenoestrogen levels, and this finding raises a safety concern.

To the best of our knowledge, the XENOBEM study is the first research to relate the total effective xenoestrogen burden, as opposed to individual xenoestrogens, to a diagnosis of leiomyomas or endometriosis. In addition, this is the first study to use omental fat for this purpose. The use of the omentum as a reference sample constitutes a novel and consistent approach, not only because adipose tissue is considered a reliable source for assessing the cumulative and long-term effects of xenoestrogens, but also because of the special metabolic impact of visceral fat [14], which has been scarcely analyzed.

The TEXB assessment protocol allows the separation of endogenous hormones, avoiding interference with the exogenous compounds contained in the alpha fraction. This is an advantage itself, but is even more important in the present study due to the presence of a high number of pregnant women in the control group. This approach has been successfully employed to investigate the relationship between xenobiotics and certain gynecological malignancies such as breast cancer [6, 16]. Other researchers have highlighted the importance of analyzing xenobiotic mixtures rather than individual compounds when establishing the potential association of xenobiotics with benign pathologies, such as uterine fibroids [25].

Finally, the XENOBEM study has an easily replicable design that, carried out on a larger scale, can be very useful in investigating the influence of xenobiotics in different pathologies.

The control group consisted predominantly of pregnant women, which might appear as a limitation of this study. Notably, the fractionation process eliminated the influence of endogenous hormones, thereby validating the design. Nevertheless, the intricate metabolic processes that occur during pregnancy can still affect the metabolism and effects of xenobiotics. Particularly in the third trimester, a state of generalized lipolysis occurs, conditioning a rapid clearance of xenobiotics and consequently, a loss of accumulated xenoestrogenicity in adipose tissue [26]. In the present study, this fact does not seem to have affected the results, as no lower levels of xenoestrogens were found in pregnant women than in the rest of the patients.

The main limitation of the XENOBEM study lies in its sample size, which is accentuated by the number of samples with TEXB-α below the limit of detection. Larger-scale studies would be beneficial in corroborating our findings.

## Conclusions

The results of the XENOBEM study confirmed the high exposure of individuals to xenoestrogens in our population. Although TEXB-α concentrations in the omentum do not seem to be associated with leiomyomas or endometriosis, the finding of detectable levels of xenoestrogens in most patients highlights the importance of regulating the use of these substances, which have proven to have harmful effects on humans and animals. The association between xenoestrogen concentrations and working or living in highly polluted areas confirms the remarkable effect of environmental pollution on the absorption of xenobiotics. The unexpected association between TEXB-α levels and estrogen-containing contraceptives warrants further investigation.

## Supporting information

**S1 Table. Main sociodemographic and clinical characteristics of controls and cases with detectable xenoestrogen levels.**
(XLSX)

**S1 Database.**
(XLSX)

**S1 File. Database encoding.**
(PDF)

## Acknowledgments

We would like to express our sincere gratitude to Marina Pollan, from the National Center for Epidemiology, for her assistance in the study design; and to Marieta Fernández, from the Centre for Biomedical Research (CBIM)-University of Granada, who directed the biochemical analyses and contributed to understanding the implication of the results regarding TEXB-α levels.

## Author Contributions

**Conceptualization:** Victoria Valdes-Devesa, David Sanz-Rosa, Ricardo Sainz de la Cuesta.

**Data curation:** Victoria Valdes-Devesa, Israel J. Thuissard-Vasallo, Cristina Andreu-Vázquez.

**Formal analysis:** Victoria Valdes-Devesa, Israel J. Thuissard-Vasallo, Cristina Andreu-Vázquez.

**Funding acquisition:** Victoria Valdes-Devesa, Ricardo Sainz de la Cuesta.

**Investigation:** Victoria Valdes-Devesa, David Sanz-Rosa, Israel J. Thuissard-Vasallo, Ricardo Sainz de la Cuesta.

**Methodology:** Victoria Valdes-Devesa, David Sanz-Rosa, Israel J. Thuissard-Vasallo, Cristina Andreu-Vázquez, Ricardo Sainz de la Cuesta.

**Project administration:** Victoria Valdes-Devesa.

**Supervision:** Victoria Valdes-Devesa, David Sanz-Rosa, Ricardo Sainz de la Cuesta.

**Validation:** Victoria Valdes-Devesa, Ricardo Sainz de la Cuesta.

**Visualization:** Victoria Valdes-Devesa.

**Writing – original draft:** Victoria Valdes-Devesa.

**Writing – review & editing:** Victoria Valdes-Devesa, David Sanz-Rosa, Israel J. Thuissard-Vasallo, Cristina Andreu-Vázquez, Ricardo Sainz de la Cuesta.

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
