## [Decision Letter · Decision Letter 0]

27 Mar 2024

PONE-D-23-42766Xenoestrogen concentration in women with endometriosis or leiomyomas: A case-control studyPLOS ONE

Dear Dr. Valdes-Devesa,

Thank you for submitting your manuscript to PLOS ONE. After careful consideration, we feel that it has merit but does not fully meet PLOS ONE’s publication criteria as it currently stands. Therefore, we invite you to submit a revised version of the manuscript that addresses the points raised during the review process.

We look forward to receiving your revised manuscript.

Kind regards,

Kazunori Nagasaka

Academic Editor

PLOS ONE

Journal Requirements:

3. We note that your Data Availability Statement is currently as follows: All relevant data are within the manuscript

Additional Editor Comments:

Dear Authors,

Thank you very much for submission to Plos One.

Our reviewers have commented on your manuscript.

Please revise the manuscript according to their comments.

I look forward to your revision soon.

Sincerely,

Kazunori Nagasaka

PLoS One

Reviewers' comments:

Reviewer's Responses to Questions

**Comments to the Author**

1. Is the manuscript technically sound, and do the data support the conclusions?

Reviewer #1: Yes

Reviewer #2: Partly

Reviewer #3: No

2. Has the statistical analysis been performed appropriately and rigorously? 

Reviewer #1: Yes

Reviewer #2: Yes

Reviewer #3: No

3. Have the authors made all data underlying the findings in their manuscript fully available?

Reviewer #1: Yes

Reviewer #2: Yes

Reviewer #3: Yes

4. Is the manuscript presented in an intelligible fashion and written in standard English?

Reviewer #1: Yes

Reviewer #2: Yes

Reviewer #3: Yes

5. Review Comments to the Author

**Reviewer #1:** The aim of this study was to examine xenoestrogen concentration differences between women with leiomyomas or endometriosis and control women and to study the relationship between the clinical and sociodemographic characteristics of these patients and their xenoestrogen levels.

Authors reported no association between omental xenoestrogen levels and leiomyomas or endometriosis but their presence, in most participants and their association with highly polluted areas, emphasizes the importance of limiting environmental exposure to these substances. An association between hormonal contraceptive use and xenoestrogen concentration was also found.

I have the following comments for the Authors:

• Please revise your English and check for typos and tense mistakes.

• Methods: Authors should state clearly the outcomes of the study, and which were the related endpoints.

**Reviewer #2:** In the manuscript entitled “Xenoestrogen concentration in women with endometriosis or leiomyomas:

A case-control study” by Valdes-Devesa et al., the Authors analyzed the possible correlation between xenoestrogen concentration and the occurrence of leiomyomas or endometriosis in a case-control study. The study demonstrated that no statistically significances between TEXB- α and the uterine pathologies. The topic is interesting but some concerns should be addressed:

1. The Authors should modify the Abstract and add a background about Endocrine disrupting chemicals (EDCs) and Xenoestrogen

2. (line 89) “….with other non hormonal benign diseases”, specify them.

3. (line 110) “….17-β-estradiol was added to the culture”, of what?

4. (line 116) “The minimum concentration of TEXB-α required to elicit a significant proliferative effect”, of what?

5. It is not clear why in the Materials and Method the Authors reported “control group included women with other non hormonal benign diseases or pregnant women who needed a cesarean section”, instead in the Statistical analysis the said “In order to homogenize the groups, we performed a comparison of xenoestrogen concentrations including only pregnant women in the control group”. Are there different control groups? It is confusing.

**Reviewer #3:** Thank you for the opportunity to review this manuscript. I have several concerns.

1. Comparison Between Pregnant and Non-Pregnant Participants

a. Mean Xenoestrogen Levels: It is crucial to address potential differences in mean xenoestrogen levels between pregnant (control) and non-pregnant (case) participants. Factors such as gestational age in pregnant women and different basic characteristics like age and BMI between the two groups could significantly impact the results. Authors should consider including a validation to compare them. Additionally, the type of surgery (laparotomy vs. laparoscopic) could also affect the outcomes and should be addressed.

2. Study Protocol Clarity

a. Study Sample Enrollment Flow: A visual representation of the study sample enrollment flow, possibly in a figure, would greatly enhance clarity. This includes the explanation of the idea that they collected only pregnant women for the control group and only non-pregnant women for the case group.

b. Xenoestrogen Measurement: More details are needed regarding the materials and methods for measuring xenoestrogens in serum. In my opinion, it was uneasy to detect which material (omental fat or serum) was used for measurements in the Methods and Results section.

3. Clinical Implications

a. Clinical Implications: The manuscript should discuss more about the clinical implications of finding higher TEXB-alpha levels in women living or working in the Madrid Community within the case group, but not in the control group.

b. Effect of Time: Consider discussing whether the eight-year gap between finishing treatment and surgery could have affected xenoestrogen concentrations. Clarification is needed regarding the rapid metabolism of estrogens from contraceptives (lines 289-291).

Minor Comments:

1. Clarify the exact number of cesarean section cases instead of stating "approximately 90%".

2. Define "Madrid Community" and whether it refers to an urban area or a specific region.

3. Results mentioned in lines 206-211 and 227-230 should be presented as figures for better comprehension.

4. Clarify whether xenoestrogen levels were higher or lower in this study compared to the ENDO study (lines 268-270).

5. Consider presenting results for the control group similar to Figure 2.

6. PLOS authors have the option to publish the peer review history of their article (what does this mean?). If published, this will include your full peer review and any attached files.

Reviewer #1: No

Reviewer #2: No

Reviewer #3: No

---

## [Author Response · Author response to Decision Letter 0]

2 May 2024

Reviewer #1: 

-Please revise your English and check for typos and tense mistakes.

We have performed the language revision

-Methods: Authors should state clearly the outcomes of the study, and which were the related endpoints. 

We have added these subheadings to the material and methods section (lines 88-93)

Reviewer #2: 

1. The Authors should modify the Abstract and add a background about Endocrine disrupting chemicals (EDCs) and Xenoestrogen. 

We have added a paragraph containing this information to the abstract section (lines 27-30).

2. (line 89) “….with other non hormonal benign diseases”, specify them: 

Tubo-ovarian abscesses secondary to pelvic inflammatory disease or patients with hemorrhagic ovarian folicles that underwent surgery (lines 98-99)

3. (line 110) “….17-β-estradiol was added to the culture”, of what? 

The e-screen bioassay tests the proliferative effect of the xenoestrogen mixtures obtained from the sample reference on a cell line extracted from breast cancers, called MCF-7. This is compared to a culture of MCF-7 cells treated with estradiol at different concentrations to calculate the prolifferative effect of the xenoestrogens contained in the sample (TEXB- α). The TEXB-α is considered a reliable biomarker for the combined effect of mixtures of xenoestrogens. We have added this information to the text (line 120).

4. (line 116) “The minimum concentration of TEXB-α required to elicit a significant proliferative effect”, of what? 

On the MCF-cell culture, we have made such clarification in the text (line 127) 

5. It is not clear why in the Materials and Method the Authors reported “control group included women with other non hormonal benign diseases or pregnant women who needed a cesarean section”, instead in the Statistical analysis the said “In order to homogenize the groups, we performed a comparison of xenoestrogen concentrations including only pregnant women in the control group”. Are there different control groups? It is confusing.

Both the case and control groups included pregnant and non-pregnant individuals. Xenoestrogen levels were compared between the case and control groups as a whole. Additionally, to mitigate the potential influence of including both types of patients in each group, we conducted an additional analysis. Pregnant women were excluded from the case group, and non-pregnant women were excluded from the control group. Consequently, this secondary analysis compared xenoestrogen levels between a case group consisting only of non-pregnant women and a control group consisting only of pregnant women. This clarification has been added to the Materials and Methods and Results sections (lines 148-152, 202-205)

Reviewer #3: 

1. Comparison Between Pregnant and Non-Pregnant Participants

a. Mean Xenoestrogen Levels: It is crucial to address potential differences in mean xenoestrogen levels between pregnant (control) and non-pregnant (case) participants. 

We initially compared xenoestrogen levels between cases and controls, both groups comprising a mixture of pregnant and non-pregnant women). Then we conducted an additional analysis after homogenizing the groups, leaving only non-pregnant women in the case group and pregnant women in the control group (results, line 148-152). This investigation confirmed the absence of differences in xenoestrogen levels between pregnant and non-pregnant women.The discussion section addressing this aspect has also been expanded (lines 362-366)

- Factors such as gestational age in pregnant women and different basic characteristics like age and BMI between the two groups could significantly impact the results. Authors should consider including a validation to compare them.

 Gestational age: As stated in the introduction (lines 57-59), xenobiotics undergo slow metabolic and detoxification processes, remaining in the body for extended periods, often several years. Therefore, the slight difference in gestational age among the cesarean section group does not significantly affect the levels of TEXB-α.

Age and BMI: As shown in Table 1, there were no significant differences in BMI between both groups. Nevertheless, age and body mass index (BMI) are factors that can modify TEXB-α levels. We analized this information, finding no relationship between age or BMI and xenoestrogen levels as mentioned in the results section, line 217. We have added a paragraph regarding this aspect in the discussion section (lines 276-296)

The type of surgery (laparotomy vs. laparoscopic) could also affect the outcomes and should be addressed.

 We used the same technique for obtaining the sample in open surgery and laparoscopy (coagulation and cutting with an electrosurgical unit, and application of bipolar energy if necessary to promote hemostasis). Therefore, the quality or composition of the sample does not change depending on the surgical approach.

2. Study Protocol Clarity

a. Study Sample Enrollment Flow: A visual representation of the study sample enrollment flow, possibly in a figure, would greatly enhance clarity. This includes the explanation of the idea that they collected only pregnant women for the control group and only non-pregnant women for the case group. 

-Such figure has been added (Fig 1). 

-As mentioned above, both groups included pregnant and non pregnant women.

-The issue regarding the composition of the case and control group has been clarified in the text (lines 162-166)

b. Xenoestrogen Measurement: More details are needed regarding the materials and methods for measuring xenoestrogens in serum. In my opinion, it was uneasy to detect which material (omental fat or serum) was used for measurements in the Methods and Results section. 

There was no measurement of xenoestrogen levels in serum. Only omental fat was analysed in this study. This has been clarified in both sections (lines 93,159).

3. Clinical Implications

a. Clinical Implications: The manuscript should discuss more about the clinical implications of finding higher TEXB-alpha levels in women living or working in the Madrid Community within the case group, but not in the control group.

We did find statistically significant differences regarding the association of contraceptive use or residence/workplace in Madrid and xenoestrogen levels. Additionally, the results of the univariate analysis were confirmed in the multivariate analysis, and an association between contraceptive use and place of residence within the Community of Madrid was ruled out (line 241-243, Fig 6). It was observed that the percentage of contraceptive users was similar between patients living inside and outside the Community of Madrid. Therefore, we can say that this was not a random finding, although we cannot provide an explanation based on the available data. However, the presence of disease differentiates both groups and other factors underneath this fact can play a role. A larger-scale study would be necessary to confirm our findings.

b. Effect of Time: Consider discussing whether the eight-year gap between finishing treatment and surgery could have affected xenoestrogen concentrations. Clarification is needed regarding the rapid metabolism of estrogens from contraceptives (lines 289-291). 

The 8-year gap supports, indeed, the validity of the finding. The estradiol contained in contraceptives behaves like endogenous estradiol and does not accumulate long-term. The finding of higher levels of xenobiotics long time after discontinuing treatment suggests the presence of another source of xenobiotics, possibly in the packaging or enteral coating of the product. Although we cannot explain the exact origin of the disruptor, various theories are proposed in the discussion. Additionally, we have modified the discussion for better understanding (lines 326-328)

Minor Comments:

1. Clarify the exact number of cesarean section cases instead of stating "approximately 90%". 

The correction has been made (lines 162-166)

2. Define "Madrid Community" and whether it refers to an urban area or a specific region. 

Such explanation is given in the discussion section, (lines 309-310)

3. Results mentioned in lines 206-211 and 227-230 should be presented as figures for better comprehension 

Lines 206-211: We have created a table , but it would be very similar to Table 1, already present in the article. We believe that including it directly in the text might impede readability. Therefore, we have provided it as supporting information (S1 Table)

Lines 227-230:The figure has been added to the text (Fig. 5)

4. Clarify whether xenoestrogen levels were higher or lower in this study compared to the ENDO study (lines 268-270). 

The ENDO study focused on analyzing the potential association between fibroids and persistent organic pollutants, examining serum and omental samples from women undergoing surgery for uterine fibroids. It shares with the XENOBEM study the use of omental fat to measure xenoestrogens. Nevertheless, whereas our study measures the overall effect of total xenoestrogen burden, the ENDO study determined the levels of 60 individual xenoestrogens concluding that these are higher in omental fat than in serum and highlighting the possible association between fibroids and specific xenoestrogens. The total burden of xenoestrogens was not quantified in the ENDO study. Therefore, such a comparison is not feasible. The text has been reformulated for better comprehension (lines 300-306).

5. Consider presenting results for the control group similar to Figure 2. 

This figure has been added (Fig 4)

---

## [Decision Letter · Decision Letter 1]

20 May 2024

Xenoestrogen concentration in women with endometriosis or leiomyomas:

A case-control study

PONE-D-23-42766R1

Dear Dr. Valdes-Devesa,

We’re pleased to inform you that your manuscript has been judged scientifically suitable for publication and will be formally accepted for publication once it meets all outstanding technical requirements.

Kind regards,

Kazunori Nagasaka

Academic Editor

PLOS ONE

Additional Editor Comments (optional):

Dear Authors,

Congratulation!

I am pleased to tell you your manucript is acceptable for publication in PLos One.

The manuscript is very interesting and useful.

We look forward to your future manuscript and please submit to our journal.

Sincerely,

Kazunori Nagasaka

Reviewers' comments:

Reviewer's Responses to Questions

**Comments to the Author**

1. If the authors have adequately addressed your comments raised in a previous round of review and you feel that this manuscript is now acceptable for publication, you may indicate that here to bypass the “Comments to the Author” section, enter your conflict of interest statement in the “Confidential to Editor” section, and submit your "Accept" recommendation.

Reviewer #1: (No Response)

Reviewer #2: All comments have been addressed

2. Is the manuscript technically sound, and do the data support the conclusions?

Reviewer #1: (No Response)

Reviewer #2: Yes

3. Has the statistical analysis been performed appropriately and rigorously? 

Reviewer #1: (No Response)

Reviewer #2: Yes

4. Have the authors made all data underlying the findings in their manuscript fully available?

Reviewer #1: (No Response)

Reviewer #2: Yes

5. Is the manuscript presented in an intelligible fashion and written in standard English?

Reviewer #1: (No Response)

Reviewer #2: Yes

6. Review Comments to the Author

Reviewer #1: I kindly thank the authors for answering to all my comments. I believe the article was greatly improved with revisions and is now suitable for publication.

Reviewer #2: I have no additional comments. Authors have addressed and answered all the Reviewer' points. It is now suitable for publication.

7. PLOS authors have the option to publish the peer review history of their article (what does this mean?). If published, this will include your full peer review and any attached files.

Reviewer #1: No

Reviewer #2: No

---

## [Editor Report · Acceptance letter]

24 May 2024

PONE-D-23-42766R1 

PLOS ONE

Dear Dr. Valdes-Devesa, 

I'm pleased to inform you that your manuscript has been deemed suitable for publication in PLOS ONE. Congratulations! Your manuscript is now being handed over to our production team.

Kind regards, 

on behalf of

Professor Kazunori Nagasaka 

Academic Editor

PLOS ONE